# Metagenomic Analysis Reveals a Mitigating Role for *Lactobacillus paracasei* and *Bifidobacterium animalis* in Experimental Periodontitis

**DOI:** 10.3390/nu14102125

**Published:** 2022-05-19

**Authors:** Fang Wu, Bing Fang, Guna Wuri, Liang Zhao, Fudong Liu, Ming Zhang

**Affiliations:** 1School of Food and Health, Beijing Technology and Business University, Beijing 100084, China; wufangleon@163.com (F.W.); wrgn1996@163.com (G.W.); 2Key Laboratory of Precision Nutrition and Food Quality, Department of Nutrition and Health, China Agricultural University, Beijing 100083, China; bingfang@cau.edu.cn; 3Key Laboratory of Functional Dairy, College of Food Science and Nutritional Engineering, China Agricultural University, Beijing 100083, China; liangzhao@cau.edu.cn; 4Beijing Laboratory of Food Quality and Safety, College of Food Science and Nutritional Engineering, China Agricultural University, Beijing 100083, China; 5Inner Mongolia Dairy Technology Research Institute Co., Ltd., Hohhot 010110, China; Liufudong@yili.com; 6Inner Mongolia Yili Industrial Group Co., Ltd., Hohhot 010110, China

**Keywords:** metagenomics analysis, periodontitis, *Lactobacillus paracasei*, *Bifidobacterium animalis*, taxonomic composition, virulence factors

## Abstract

Probiotics have aroused increasing concern as an intervention strategy for periodontitis (PD), but their underlying mechanism of action remains poorly characterized. Regarding the significance of oral microbiota dysbiosis related to PD, we predicted that the preventive activity of probiotics may be influenced by suppressing the bacterial pathogenicity. Herein, we investigated the effects of *Lactobacillus paracasei* L9 (L9) and *Bifidobacterium animalis* A6 (A6) on PD using a rat model, and demonstrated a regulatory effect of probiotics on oral flora from a metagenomics perspective. Oral administration of A6 or L9 effectively relieved gingival bleeding, periodontal inflammatory infiltration, and alveolar bone resorption. In addition, A6 or L9 treatment reduced the inflammatory response and increased the expression of anti-inflammatory cytokines, which we expected to ameliorate alveolar bone resorption as mediated by the receptor activator of the nuclear factor-κB ligand/OPG signaling pathway. More importantly, using metagenomic sequencing, we showed that probiotics significantly altered the taxonomic composition of the subgingival microbiome, and reduced the relative proportions of pathogenic bacterial genera such as *Streptococcus*, *Fusobacterium*, *Veillonella*, and *Escherichia*. Both probiotics significantly inhibited levels of bacterial virulence factors related to adherence, invasion, exoenzyme, and complement protease functions that are strongly correlated with the pathogenesis of PD. Our overall results suggest that A6 and L9 may constitute promising prophylactic agents for PD, and should thus be further explored in the future.

## 1. Introduction

Periodontitis (PD) is a microbe-mediated oral disease that destroys periodontal tissues that include gingiva, periodontal ligaments, and alveolar bone, which subsequently results in tooth movement or loss [1]. Statistically, the prevalence of PD among middle-aged and older individuals in China has reached 70% [2]. Severe PD is the sixth most prevalent disease worldwide, with a total incidence rate of 11.2% [3]. A global epidemiological survey of oral health revealed that periodontal disease constitutes a significant factor that necessitates tooth extraction or causes tooth loss [4]. Moreover, PD also presents a huge economic burden to patients. The authors of one survey found that the average total treatment and annual treatment costs for patients with PD were EUR 7154 and EUR 437, respectively [5]. Routine treatment of PD includes nonsurgical treatment (subgingival curettage) combined with antibiotics [6], and aims to remove pathogenic plaque biofilms and tooth calculus burden through mechanical procedures [7]; however, periodontal pathogens can regroup, and mechanical surgery thereby only suffices as a short-term treatment.

A large number of pathogenic bacteria, such as *Porphyromonas gingivalis* (*P. gingivalis*), proliferate under the gingiva to form biofilms, and mature plaque biofilms adhere to one another or adhere to the surface of teeth, between teeth, or the surface of a prosthesis [8]. These bacteria, then, use their own virulence factors to destroy and infect periodontal tissue, gradually resulting in inflammatory destruction of periodontal tissue and the loss of alveolar bone [9]. In a landmark study, investigators identified five major bacterial complexes in subgingival plaque, particularly the “red-complex”, which consists of *P. gingivalis*, *Tannerella forsythia*, and *Treponema denticola*. These three pathogens tend to associate and show close involvement with periodontal disease pathogenesis [10]. However, technological developments in next-generation sequencing have expanded researchers’ understanding of oral microecology. One study suggests that over 700 bacterial taxa inhabit the human oral cavity, and that some overlooked species are also critical to the underlying process of PD [11]. These studies provide a new perspective on the prevention of periodontal disease and the strategies to be used for controlling pathogenic bacteria. Similar to aspects of gut microflora, oral microflora may constitute a novel target for the prevention of PD.

Recent studies have suggested that probiotics potentially offer a means to prevent and treat PD. Matsubara et al. summarized 12 randomized controlled trials (RCTs) on the interventional effects of probiotics on PD and concluded that oral administration of a certain daily content of probiotics as the only means or auxiliary means of PD prevention and treatment significantly improved the typical symptoms manifested by patients with PD [12]. Another clinical finding suggested that intervention with heat-killed *Lactobacillus plantarum* L-137 could inhibit the deepening of periodontal pockets [13], and the adjunctive use of *Lactobacillus reuteri* and systemic antibiotics showed similar improvements in all clinical periodontal parameters [14]. The improvement in clinical symptoms may be due to changes in the periodontal microenvironment. One study detailed the use of *Bifidobacterium animalis* subsp. lactis HN019 as an adjunct to clinical therapy in the treatment of chronic PD [15], while another found that oral administration of *Lactobacillus salivarius* diminished the numbers of periodontopathic bacteria in plaque samples [16]. Although the majority of the literature predicts that probiotics treatment may reconstruct the health-related oral microbiome and thus alleviate the symptoms of PD, the relevant underlying mechanism or mechanisms have not been verified further.

Our previous study revealed that *Lactobacillus paracasei* L9 (L9) robustly inhibited *Streptococcus mutans* biofilm formation in vitro [17], and another showed that *Bifidobacterium animalis* subsp. lactis A6 (A6) as a probiotic offered high acid resistance and improved inflammatory response [18,19]. The aim of the current study, then, was to investigate the ameliorating effect of A6 and L9 on a ligature-induced PD rat model, and to further verify their modifying effects on pro-inflammatory responses and the expression levels of key biomarkers in the experimental PD rat. In addition, we wished to assess whether A6 and L9 application altered taxonomic composition of the oral microbiome using metagenomic sequencing. *Streptococcus salivarius* strain K12 (K12, commercial oral probiotics) was used as the reference strain.

## 2. Materials and Methods

### 2.1. Animals and Groups

Fifty male Wistar rats (6–8 weeks of age, SPF grade, and 200–220 g) were obtained from HFK Bioscience Co. (Beijing, China). All rats were maintained in standard cages and under identical conditions (including a 12 h/12 h light-dark cycle) with access to drinking water and food ad libitum. All protocols and procedures followed the National Research Council’s Guide for the Care and Use of Laboratory Animals, and were approved by the Ethics Committee of China Agricultural University (AW01210202-4-2).

### 2.2. Bacterial Strains

*P. gingivalis* W83 was grown on blood agar medium supplemented with menadione (1 μg/mL) and hemin (5 μg/mL, Sigma-Aldrich, St. Louis, MO, USA) and maintained in an anaerobic environment (AnaeroPack-Anaero, Mitsubishi Gas Chemical Co., Tokyo, Japan) at 37 °C for 3–5 days. L9 (China General Microbiological Culture Collection (CGMCC) 3606) was isolated from fecal samples of healthy centenarians and cultured in MRS broth at 37 °C, and A6 (CGMCC 9273) was provided by the Functional Dairy Laboratory of China Agricultural University, and anaerobically cultured in MRS medium at 37 °C. K12 cells were cultured in standard M17 medium at 37 °C.

### 2.3. Experimental Design

Before the experiment was initiated, animals were acclimated to an animal room for seven days. The 50 rats were split equally into five groups: Group 1, untreated controls (Control); Group 2, the PD model (Model); Group 3, A6 (1 × 10^8^ CFU/mL, 20 mL/d) treatment after PD ligature (A6); Group 4, L9 (1 × 10^8^ CFU/mL, 20 mL/d) treatment after PD ligature (L9); and Group 5, K12 (1 × 10^8^ CFU/mL, 20 mL/d) treatment after PD ligature (K12). Groups 3–5 were administered probiotics in drinking water from the 2nd to 8th week, with the probiotics replenished every 12 h.

After eight weeks, we diagnosed our clinical indicators and collected serum samples from rats for cytokine analysis. The rats were then sacrificed, and we preserved the maxillary periodontal tissue in 4% paraformaldehyde for subsequent micro-CT observation and histopathological analysis. Experimental animals underwent general anesthesia with 7% chloral hydrate (0.5 mL/100 g), and we established our PD model by placing sterile nonabsorbent surgical suture on both sides of the maxillary second molar and knotting it in the proximal location [20]. Silk ligature was immersed in 1 × 10^9^ CFU of *P. gingivalis* in 100 µL of PBS with 2% carboxymethylcellulose (CMC), and we applied the bacterial liquid twice a week [21]. The ligature was kept in a fixed position in all animals throughout the experiment [22].

### 2.4. Clinical Indicators and Histological Assessment

Gingival inflammation was evaluated prior to rat sacrifice in each group, and we selected the following four clinical indicators according to the literature: gingival index (GI), sulcus bleeding index (SBI), bleeding on probing (BOP), and probing depth (PD) [23,24].

The teeth and surrounding soft tissue were fixed in 4% paraformaldehyde, and post-fixation they were removed and rinsed under running water for 2–3 h and decalcified by transferring to a 15% EDTA decalcification solution for 50 days. The periodontal tissues were then cut into 1 cm × 1 cm-sized tissue blocks, dehydrated in an increasing ethanol gradient, and immersed in liquid paraffin overnight. After the embedded wax tissue blocks solidified, continuous 5 μm sections were cut and representative sections were selected for analysis using hematoxylin and eosin (H&E) [25].

### 2.5. Alveolar Bone Loss Assessment by Micro-CT Analysis

After removing the soft tissue from the right maxilla, we fixed it in 4% paraformaldehyde solution, and the paraformaldehyde-fixed maxillae were scanned with an Inveon MM CT (SIEMENS, Munich, Germany). We exploited the Inveon Research Workplace microcomputer image processing system to analyze the bone histomorphology and alveolar bone, and linearly determined the loss of alveolar bone from the cementoenamel junction (CEJ) to the alveolar bone crest (ABC) as described previously [26]. The measurements were performed three times per site in a blinded manner.

### 2.6. Reverse Transcription Polymerase Chain Reaction (RT-PCR) Analysis

Gingival tissues were first carefully separated and homogenized using a tissue grinder, and SYBR Green I real-time PCR was employed to detect the changes in the mRNA transcription level of a target gene. cDNA synthesis was performed with a PrimeScript™ RT reagent Kit with a gDNA Eraser (TaKaRa, Catalog Number RR047B). The target gene and internal reference for each sample were subjected to RT-PCR amplification, each sample was tested in three replicate wells, and the data were analyzed using the 2^−^^ΔΔCT^ method. Primers were synthesized by Invitrogen (Beijing, China), and their sequences are listed in Table 1.

### 2.7. Enzyme-Linked Immunosorbent Assay (ELISA) of Serum

Blood was collected from rats after an overnight fast at the termination of the experiment, and serum was separated by centrifugation at 3000 rpm for 15 min and stored at −80 °C until required. Serum samples were analyzed for the presence of rat homologs of interleukin-10 (IL-10), tumor necrosis factor-α (TNF-α), and interleukin-6 (IL-6) using a Luminex^®^ Assay Mouse Premixed Multi-Analyte Kit (Inc R&D Systems, Minneapolis, MN, USA, cat. No. LXSAMSM). All standards and samples were assayed using a Bio-Rad^®^ Bio-Plex^®^ 200 machine (Luminex Corporation, Austin, TX, USA).

### 2.8. Western Immunoblotting (WB) Analysis

Protein levels of matrix metalloprotease (MMP) and receptor activator of nuclear factor-κB ligand (RANKL) were assessed by WB analysis. The gingival tissue around the maxillary second molar was carefully removed and homogenized. After quantitative determination of protein concentration with the tissue BCA method, we executed WB analysis of the target protein and internal reference protein as previously described [27]. The exposed film was scanned directly, and the image format was converted with Image J software (NIH, Bethesda, MD, USA). The integrated optical density (IOD) values of the bands were read and analyzed with Total Lab Quant V11.5 (Newcastle upon Tyne, UK). Primary antibodies against β-actin, MMP-8, MMP-9, and RANKL were purchased from Abcam (Cambridge, MA, USA).

### 2.9. DNA Extraction, Library Construction, Metagenomic Sequencing, and Gene Taxonomic and Functional Annotation

We collected the treated ligature silk and scraped the subgingival plaque along the gingival edge of the tested teeth to complete our sample collection. An E.Z.N.A.^®^ Soil DNA Kit (Omega Biotek, Norcross, GA, USA) was used to extract total genomic DNA according to the manufacturer’s instructions, and the concentration and purity of the extracted DNA were determined with TBS-380 and a NanoDrop2000, respectively. DNA extract quality was evaluated on 1% agarose gel, and the extract was fragmented into an average size of approximately 400 bp using Covaris M220 (Gene Company Limited, Hong Kong, China) for library construction. We constructed DNA libraries using a NEXTFLEX Rapid DNA-Seq machine (BioScientific, Austin, TX, USA), while paired-end sequencing was performed on an Illumina NovaSeq (Illumina Inc., San Diego, CA, USA) at Majorbio Bio-Pharm Technology Co., Ltd. (Shanghai, China), using NovaSeq Reagent Kits according to the manufacturer’s instructions (www.illumina.com, accessed on 2 January 2022).

After quantification, the libraries were sequenced on an Illumina NovaSeq platform to obtain raw data, and clean high-quality data were obtained after filtering by fastp tools [28] (https://github.com/OpenGene/fastp, version 0.20.0, accessed on 2 January 2022). Contig libraries were assembled with MEGAHIT [29] (https://github.com/voutcn/megahit, version 1.1.2, accessed on 1 February 2022), which makes use of succinct de Bruijn graphs. Contigs with lengths over 300 bp were selected as the final assembly result, and the contigs were then used for further gene prediction and annotation.

To establish a nonredundant subgingival plaque catalog, the predicted genes were aligned using CD-HIT (identity > 95% and coverage > 90%; (http://www.bioinformatics.org/cd-hit/, version 4.6.1 and http://soap.genomics.org.cn/, version 2.21, accessed on 1 March 2022), and gene abundances in each sample were evaluated. For taxonomic and functional assignation, gene catalogs were aligned against the NCBI NR (http://blast.ncbi.nlm.nih.gov/Blast.cgi, accessed on 1 March 2022) and VFDB databases (http://www.mgc.ac.cn/VFs/, accessed on 1 March 2022) with an e-value cutoff of 1 × 10^−5^. The correlation heatmap of clinical indicators and species virulence factors was calculated using the vegan package for R.

### 2.10. Statistical Analysis

We employed GraphPad Prism 7 software for our statistical analyses. One-way ANOVA and Fisher’s exact probability test were executed for statistical comparisons, and *p* < 0.05 indicated a statistically significant difference. The metagenomics analysis was performed on the Majorbio Cloud Platform (www.majorbio.com, accessed on 20 March 2022).

The analytical methods we employed for metagenomic species, functional, and correlation analyses were as follows. The sample-species or function Circos diagram is a visual circle diagram describing the correspondence among samples, species, and functions and the distribution ratio between pairs as reflected by Circos-0.67-7 software (http://circos.ca/, accessed on 10 March 2022). PCoA (principal coordinates analysis) is a visualization method used to study the similarities or differences in a data set. Based on the Bray–Curtis distance, the potential principal components that affect the composition or functional differences of sample communities can be found through dimensionality reduction. Our data were analyzed using R-language software for PCoA, and we generated a PCoA graph. LEfSe difference discriminant analysis was used to uncover the species or functional characteristics that can best explain the differences in two groups, and the degree of influence of these characteristics on inter-group differences. Specifically, the nonparametric Kruskal–Wallis (KW) rank-sum test and Wilcoxon’s signed-rank test were initially used to test the difference in species or function abundances between different groups, and linear discriminant analysis (LDA) was executed to estimate these different species or functions and the size of the effect on the differences between groups. The analysis software we used was LEfSe (http://huttenhower.sph.harvard.edu/galaxy/root?tool_id=lefse_upload, accessed on 1 March 2022), and according to the taxonomic composition, we used LDA on the samples with different grouping conditions to ascertain the species or functions that exhibited a significant difference in the classification of the samples [30]. Species and functional differences between the two groups were analyzed with the Wilcoxon rank-sum test (a nonparametric test method for two independent samples) using the R stats and Python *scipy* packages. By studying the average rank of the two groups of samples, it was possible to judge whether there was a difference in the distribution of the two populations. With this analysis, we could implement a significant difference analysis on the species/functions of the two groups of samples, and undertake various methods of correction for the *p* value. For correlation heatmap analysis, we used R (pheatmap package) to calculate Spearman’s rank correlation coefficient between the clinical index and the selected function, and the obtained numerical matrix was visually displayed through the heatmap chart. The color depth indicated the size of the data value, and the data information on relevant degree was reflected in the color change. When four functions (adherence, invasion, exoenzyme, and complement protease) that were highly correlated with clinical index were selected for further analysis, we specifically identified the top 10 virulence factors assigned to the four functions; however, the number of exoenzyme and complement protease function virulence factors was less than 10, and we therefore listed all virulence factors in the two functions. We then exploited the GraphPad Prism 7 software to analyze the differences between the intervention groups and the model group and to construct the heatmap.

## 3. Results

### 3.1. Clinical Index Score and Alveolar Bone Aspiration

Experimental arrangement and the major pathological index were shown in Figure 1. The weights of mice increased normally throughout the entire experiment, and we uncovered no adverse reactions. Five mice were excluded due to incidental death during anesthesia or ligature. The rats in the model group exhibited typical PD-like characteristics at the termination of the experiment, including gingival bleeding and periodontal pocket deepening.

BOP and SBI are the clinical indicators of bleeding frequency and severity, and, as shown in Figure 1B, all rats in the model group (10/10) showed severe gingival bleeding. A6, L9, and K12 intervention reduced the frequency of bleeding, with a significant reduction observed for the K12 group (*p* < 0.05). All three probiotics also significantly reduced the SBI (*p* < 0.05) (Figure 1B). The depth of PD was assessed by measuring periodontal pocket depth, and as shown in Figure 1B, the A6 and L9 intervention groups showed a tendency to be reduced, although this was not statistically significant (*p* > 0.05). GI is the comprehensive score of PD symptoms that includes BOP, the SBI, and the depth of the periodontal pocket, and scores for the L9 and K12 groups were diminished from 2.22 to 1.56 and 1.44, respectively (*p* < 0.05, Figure 1B). In general, these results indicate that oral administration of the three probiotics alleviated gingival bleeding symptoms.

We determined alveolar bone aspiration via micro-CT by measuring the distance from the CEJ to the ABC. As shown in Figure 1C, the CEJ was very close to the ABC, and we observed very little alveolar bone loss in the control group; however, in the model group, the alveolar bone of the maxillary second molar absorbed up to 1/3 of the root tip, and resorption was as high as 0.66 mm. The significant variation in the model group therefore indicated that the PD model was successfully implemented. The application of L9 significantly inhibited alveolar bone loss compared to the model group (*p* < 0.05), while A6 treatment also alleviated alveolar bone loss (although the variation was not significant, *p* > 0.05).

### 3.2. Histopathological Observations

H&E staining was used to detect the pathological damage to the periodontal tissue—including inflammatory infiltration, collagen fiber rearrangement, and alveolar bone resorption. As shown in Figure 2, the integrity of the epithelium remained intact, collagen fibers and fibroblasts were regularly arranged in the periodontal tissue, and no inflammatory cells were present in the control group. In contradistinction, in the model group, epithelial proliferation formed a mesh shape, and we noted marked inflammatory infiltration throughout the connective tissue region. A6, L9, and K12 interventions significantly attenuated the degree of inflammatory cell infiltration relative to the model group, but they did not improve the proliferative capability of the epithelial roots. The epithelial injury to the interproximal area was notably mitigated by probiotic intervention, and we observed alveolar bone resorption along the distance from the CEJ to the ABC. While the model group showed considerable alveolar bone resorption, the CEJ/ABC inter-distance was significantly reduced, and alveolar bone resorption was lessened after probiotic intervention, which is consistent with the micro-CT results.

### 3.3. Probiotics Downregulate Inflammatory Cytokines in the Serum

To better understand the systemic inflammatory response in PD or to probiotics, we examined serum pro- and anti-inflammatory factors in rats by ELISA and demonstrated marked upregulation in the expression of the inflammatory factors IL-6 and TNF-α (Figure 3A,C). Compared with the model group, L9 and K12 decreased serum IL-6 and TNF-α protein levels, while in contrast, oral administration of the three probiotics significantly reversed the decline in IL-10 expression (Figure 3B, *p* < 0.05) that was induced by ligature and *P. gingivalis*.

### 3.4. Probiotics Modulate the Expression of Pro- and Anti-Inflammatory Cytokines in Gingival Tissues

Inflammatory cytokine expression in gingival tissues is an important indicator of periodontal disease progression [31]. In the present study, we analyzed inflammatory factors in gingival tissues by RT-PCR, and our data revealed that compared with the control group, the levels of TNF-α and IL-17 were notably augmented in the model group (*p* < 0.01, Figure 3D,F), while oral administration of A6 and L9 significantly reduced TNF-α mRNA levels (Figure 3D, *p* < 0.05). Furthermore, A6 and K12 also attenuated the overexpression of IL-17 (*p* < 0.05).

IL-10 is a prominent anti-inflammatory cytokine that can relieve PD by inhibiting key regulators of osteoclasts [32]. As shown in Figure 3E, although the levels of IL-10 in the model group declined significantly (*p* < 0.001), the reduced IL-10 levels were reversed after administering either L9 or K12 (*p* < 0.05), and this result is consistent with the trend for IL-10 levels in the serum noted above. These results suggest that the administration of A6 or L9 partially alleviated systemic and gingival inflammatory responses induced by PD, favored the expression of the anti-inflammatory cytokine IL-10, and inhibited the expression of pro-inflammatory cytokines.

### 3.5. Effects of Probiotics on MMP-8 and MMP-9 Protein Expression Levels in Gingival Tissues

The MMPs comprise a family of neutral endopeptidases that are widely distributed and able to degrade connective tissue [33,34]. The relative expression of MMP-8 and MMP-9 showed a tendency to be slightly elevated (but not to a significant degree) in the PD group compared with normal controls (Figure 4B,D, *p* > 0.05), and L9 and A6 treatment then decreased the relative expression of MMP-8 and MMP-9 slightly, although this was also not significant (Figure 4B,D, *p* > 0.05). In addition, when we further validated these findings by immunohistochemical staining, we observed no differences after probiotic intervention (data not shown).

### 3.6. Effects of Probiotics on Expression Levels of RANKL/OPG in Gingival Tissues

RANKL is a key regulator secreted by osteoblasts which promotes osteoclast activation and differentiation and eventually leads to the loss of alveolar bone [35]. As shown in Figure 4A, we detected robust positive expression for RANKL in the PD group compared with normal controls (*p* < 0.05), and oral administration of the three probiotics downregulated RANKL expression, with the A6 intervention group delivering the greatest diminution (Figure 4A, *p* < 0.05)

We examined OPG (a vital osteoclast inhibitor [36]) in gingival tissues by RT-PCR, and data reveal that its mRNA levels in PD rats declined significantly (*p* < 0.05, Figure 4C). Oral treatment of the three probiotics, however, upregulated the expression of OPG, while L9 treatment markedly elevated expression in gingival tissues (*p* < 0.05).

### 3.7. Probiotics Alter the Taxonomic Composition of the Subgingival Microbiome

To investigate whether subgingival probiotic bacteria mitigated PD, we executed metagenomic gene sequencing of subgingival plaque. After quality-control optimization, 1888.2 million high-quality reads with an average reading length of 133,293,513 bp were obtained. The number of optimized reads accounted for 97.28% of the original number, thus ensuring that all microorganisms in the samples were accounted for and met our analysis requirement. We then used the sequencing data to assemble a gene catalogue of 163,733 nonredundant genes, and to reveal the taxonomic composition of the periodontal flora, these genes were searched against the NCBI nonredundant protein database using BLASTx. Finally, we identified 22 phyla, 39 classes, 64 orders, 110 families, and 207 genera. Based on phylum, Actinobacteria was dominant and followed by Firmicutes, Bacteroidetes, Proteobacteria, and Fusobacteria. Based upon genus level, *Bifidobacterium* was dominant and followed by *Veillonella*, *Bacteroides*, *Streptococcus*, *Rodentibacter*, *Lactobacillus*, *Phocaeicola,* and *Fusobacterium* (Figure 5A).

Statistical analysis of similarity (ANOSIM) demonstrated that the taxonomic composition was significantly different after L9 and A6 intervention compared with the model group (*p* < 0.05). We further performed PCoA and LEfSe analyses among the three groups to investigate periodontal microbiotal differences after L9 and A6 intervention, and our PCoA depicted a distinct distribution with respect to distances for the three groups (Figure 5B), implying a significant difference in bacterial community composition. When we used LEfSe analysis to estimate species effect size per group with regard to differential effects, we identified 10 characteristic taxa at the genus level and observed that *Desulfovibrionaceae*, *Coriobacteriia,* and *Ellagibacter* were discriminative for the model group; *Olsenella*, *Enterococcus,* and *Bifidobacterium* were discriminative for the L9 group; and *Bifidobacterium*, *unclassified_d__Bacteria, Ralstonia,* and *Motilibacter* were discriminative for the A6 group (LDAs > 2.5) (Figure 5C).

When we further analyzed the abundance changes of the characteristic taxa after probiotic intervention at the genus level, we noted differences among the three characteristic strains derived from LEfSe analysis (Figure 6A). We specifically discerned that the relative abundance of *Desulfovibrionaceae* in the A6 and L9 groups was diminished, the abundance for the A6 group was significantly lower than in the model group (*p* < 0.01), and that the relative abundance of *Coriobacteriia* was significantly reduced after L9 intervention (*p* < 0.01). Figure 6B,C show the differential species using the Wilcoxon-Mann-Whitney U test, and among the bacterial genera, we observed a greater abundance of *Bifidobacterium* after oral administration of A6 compared with the PD group (*p* < 0.01). In contrast, the abundances *of Veillonella*, *Streptococcus, Rodentibacter, Fusobacterium, Desulfovibrionaceae,* and *Escherichia* were distinctly attenuated after A6 intervention (Figure 6B). In the L9 group, the abundances of *Alistipes* and *Escherichia* decreased, while the abundance of *Olsenella* significantly increased (Figure 6C).

### 3.8. Differences in Virulence Factor Expression after Probiotic Intervention

Various virulence factors (VF) from pathogenic bacteria drive the periodontal inflammatory response and gradually lead to excessive inflammation around the gingival tissue and to alveolar bone absorption. To further explore whether probiotics can improve periodontal virulence factors in rats with PD, we used the Virulence Factor Database (VFDB) (http://www.mgc.ac.cn/VFs/, accessed on 20 March 2022) to identify potential VFs by sequence alignment [37], and VF function was annotated by Diamond against the VFDB database with an e-value cutoff of 1 × 10^−5^ [38]. We ultimately identified 202 VFs and classified them by 12 aspects of function that included iron uptake system, adherence, anti-phagocytosis, secretion system, and toxin.

Among the 12 functions, the functions of adherence, invasion, exoenzyme, complement protease, and serum resistance changed significantly in the A6 group compared with the PD group (*p* < 0.05 by analysis with the Kruskal-Wallis test) (Figure 7B). In the L9 group, we noted significant differences in iron uptake system, adherence, invasion, exoenzyme, and complement protease (*p* < 0.05). Both of our probiotics significantly reduced adherence, invasion, exoenzyme, and complement protease virulence factors.

We investigated whether certain virulence factors were associated with PD index (including BOP, SBI, PD, and GI (Figure 7C)), and ascertained that virulence functions related to adherence, complement protease, exoenzyme, invasion, iron uptake system, phase variation, and toxin were positively correlated with the probing index. Four virulence functions (including adherence, invasion, exoenzyme, and complement protease) were significantly correlated with GI, and these were also significantly reduced by L9 and A6 interventions (*p* < 0.05).

We subsequently identified differences among the top 10 virulence factors assigned to adherence, invasion, exoenzyme, and complement protease function, and demonstrated that a majority of these factors (23/33) were enriched in the model groups. The virulence factors related to type-1 pili (VF0082), capsule (VF0323), BoAa (VF0434), BoAb (VF0435), flagella (VF0394), flagella (VF0114), flagella (VF0430), P60 (VF0068), LpeA (VF0346), OmpA (VF0236), type-1 fimbriae (VF221), EspC (VF0173), Tsh (VF0233), and DNase (VF0252) were curtailed in the A6 group (*p* < 0.05), while virulence factors related to BoAa (VF0434), BoAb (VF0435), flagella (VF0114), type-1 fimbriae (VF221), DNase VF0252, and sialidase (VF 0391) were diminished in the L9 group (*p* < 0.05).

## 4. Discussion

Although probiotics are suggested to provide an effective and safe contribution to the management of chronic PD, additional studies are required to explore their precise mechanisms of action. Three such predicted mechanisms currently include modulation of the oral microbiome, inhibition of pro-inflammatory responses, and the production of antimicrobial substances. We herein established an animal model of PD and found that probiotics A6 and L9 relieved gingival bleeding and alveolar bone resorption, and reduced pro-inflammatory response. More importantly, and using metagenomic sequencing, we demonstrated that oral probiotic administration significantly altered the taxonomic composition of the subgingival microbiome and reduced the virulence of pathogenic bacteria, providing a basis for the development of probiotics as interventional strategies used to prevent PD. To the best of our knowledge, the present study was the first ever to reveal a modulatory effect of probiotics on the subgingival microbiome in the experimental PD rat from a metagenomics perspective.

The formation of pathogenic microbial communities constitutes an important cause of chronic PD, and continuously improving sequencing technologies have greatly elucidated the complexity and diversity of oral microbiota. Furthermore, the structure of the oral microbiota in PD is stable, as the keystone pathogens can recover or even aggravate conditions in the mouth after tooth scaling and root planing [31]. However, whether probiotics can alter oral microbial communities remains clinically controversial [39]. Numerous in vitro studies showed that many probiotics—including *L. paracasei* and *B. animalis*—compete with *P. gingivalis* [40,41], and authors have thereby speculated a potential for integrating them into subgingival biofilms. One research team from the University of São Paulo created a checkerboard DNA-DNA hybridization technique, and demonstrated that exogenous *B. animalis* HN019 translocated into the subgingival biofilm and reduced the proportions of red and orange complexes in rats and humans [42]. Our study similarly proved that *B. animalis* A6 altered overall taxonomic composition and enhanced the abundance of Bifidobacteriaceae. In addition, the administration of *L. paracasei* L9 modulated the structure of the microbiota and, simultaneously, increased the abundance of Bifidobacteriaceae. Although a precise role for *Bifidobacterium* in the oral cavity is controversial, the antagonistic relationship between *B. animalis* and keystone pathogens appears clear in PD [15]. Therefore, the modulation of the subgingival flora by A6 and L9 and the enhancement of *Bifidobacterium* augur well for beneficial effects.

Moreover, the use of the pairwise *t* test showed that the abundances *of Streptococcus, Fusobacterium*, *Veillonella,* and *Escherichia* were notably reduced after A6 intervention (Figure 6B), with *Streptococcus* and *Fusobacterium* known to be common inhabitants of the human oral cavity. Some species of *Fusobacterium* also directly invade host periodontal cells and induce the production of pro-inflammatory factors [43]. *Veillonella* species, as early colonizers of oral bacteria, occupy critical roles in constructing multispecies dental plaque [44,45]. While the abundances of *Alistipes* and *Escherichia* in the L9 group decreased significantly (Figure 6C), their roles during PD continue to be obscure. *Alistipes* was found to be associated with PD, and the exopolysaccharides of *Escherichia* contributed to its pathogenicity in gingival epithelial cells [46,47]. Collectively, administration of A6 and L9 considerably reduced the relative proportions of typical pathogenic bacteria.

Metagenomic sequencing analyses provide vast amounts of information regarding the oral microbial community and the shift to low-abundance species in particular. For example, Ng et al. concluded that species in low abundance acted as critical species during the pathogenesis of PD [48]. We performed LEfSe analysis and revealed that Desulfovibrionaceae and Coriobacteriia contained genera critical to the model group. Desulfovibrionaceae, as members of the sulfate-reducing bacteria, produce H_2_S, a virulence factor that gradually results in inflammation and irreversible degradation of gingival tissue [49]. Coriobacteriia have been shown to be strongly associated with aggressive PD [50], and A6 and L9 intervention significantly reduced the abundance of both critical pathogens.

An advantage of the shotgun metagenome-sequencing approach was that it allowed us to predict functional genes associated with PD, and among these genes, bacterial virulence factors have been suggested to be initial elements during PD pathogenesis [51]. In our study, both of the probiotics that we used significantly reduced the virulence factors associated with adherence, invasion, exoenzyme, and complement protease functions—all of which were all strongly correlated with GI. Adherence to and invasion of the host cell are necessary prerequisites for bacterial infection [51]. In our study, five adherence factors were reduced by A6 intervention, including type-1 pili and capsule. Type-1 pili control the maneuverability and chemotaxis of potentially pathogenic bacteria and guide the microorganism to allow them to search new sites for colonizing or damaging the target tissues, resulting in improved survival in a complex oral–flora environment [52]. Functionality–capsule synthesis can improve the survival and persistence of bacteria in an unfavorable environment and promote inflammatory reactions [53]. The three adhesins IlpA, BoaA, and BoaB were inhibited by both A6 and L9; in addition, two invasive factors were also reduced by both A6 and L9 intervention, including flagella and type-1 fimbriae. The flagella factor was thought to be related to motility and the environmental-sensing functions of species such as *Pseudomonas aeruginosa* and *Neisseria gonorrhoeae* [54]. Fimbriae of oral pathogens also possess filamentous components on their cell surfaces and are regarded as critical to host immune responses and the invasion of periodontal tissues [55]. We noted that L9 significantly reduced the levels of exoenzyme genes, particularly sialidase, and sialidase constitutes another adverse factor distributed by oral pathogens such as *Tannerella forsythia* [56]. Bacterial sialidase also cleaves terminal sialic acid residues from carbohydrate polymers and thus further enhances bacterial attachment or tissue destruction [57]. Overall, our findings reveal that dietary A6 or L9 not only modulated subgingival microflora but also repressed the levels of bacterial virulence factors based on microbiotal metagenomic sequencing.

Bone loss is one of the most serious symptoms of PD, and previous studies have shown that RANKL/OPG pathways play vital roles during alveolar bone aspiration [58]. While upregulation of RANKL expression stimulated osteoclast differentiation and eventually led to alveolar bone resorption [59], OPG secreted by osteoblasts was primarily efficiently bound to RANKL and protected the skeleton from excessive bone resorption [36]. Previous experiments on the prevention and treatment of PD showed that the downregulation of RANKL and other factors was closely related to the relief of periodontal symptoms [60,61]. In addition, in our experiment, the expression of RANKL was downregulated by A6 treatment, while OPG was upregulated by L9 treatment. H&E and micro-CT results also reveal that L9 intervention suppressed alveolar bone resorption. These data thus suggest that bone loss that mitigated the effects of probiotics correlated with modulation of the OPG/RANKL signaling pathway.

The bone loss and high expression of RANKL in the pathogenesis of PD are related to overexpression of inflammatory factors, which can aggravate the expression of RANKL. A previous study showed that oral administration of probiotics effectively relieved alveolar bone absorption and downregulated several pro-inflammatory cytokines [62]. Investigators also found that IL-17 not only enhanced the expression of RANKL in osteoblasts via T cells [63] but also elevated TNF-α and other inflammatory factors so as to promote RANKL signal transduction [64], stimulate osteoclast maturation, and ultimately cause the progressive absorption of alveolar bone [65]. Periodontal inflammatory TNF-α is mainly generated by the reaction of macrophages to lipopolysaccharide and also promotes the bodily production of collagenase and the absorption of bone tissue [33]. In the current study, A6 or L9 treatment suppressed pro-inflammatory cytokines and stimulated IL-10 expression in both serum and gingival tissues, and may constitute one of the mechanisms that underlies the prevention and treatment of PD by probiotics.

We expect that the results from the present study will provide evidence for the adjuvant treatment of PD with probiotics and provide new alternative bacterial strains for oral healthcare products, such as toothpastes, mouthwash, oral sprays, and other daily chemical products, or for yogurts and other foods with oral health functions. This study was an exploration of animal model of PD, which inevitably has some limitations. Although the pathogenetic process of the rat ligature model is similar to human PD, the structure and function of rat teeth are somewhat different from those of humans. Moreover, due to the effect of heredity, diet, and environment, the oral microbial structure of rats is quite different from that of humans. Therefore, the application of experimental bacterial strains to humans requires additional research.

## 5. Conclusions

In summary, the results from the current study indicate that oral administration of A6 or L9 effectively relieved the clinical symptoms of PD, reduced inflammatory responses, and increased the expression of anti-inflammatory cytokines. We expect that this mitigation would further alleviate alveolar bone resorption mediated by the RANKL/OPG-signal-transduction pathway. Importantly, using metagenomic sequencing, we herein demonstrated that oral administration of A6 or L9 significantly altered the taxonomic composition of the subgingival microbiome, reduced the relative proportions of typical pathogenic bacteria, and downregulated the virulence of pathogenic bacteria. To the best of our knowledge, the current study was the first ever to illuminate the effects of probiotics on the subgingival microbiome from a metagenomics perspective in the experimental PD rat. Our findings provide new evidence for the prevention of periodontal disease by probiotics, and we suggest that A6 and L9 be further explored as promising prophylactic agents for PD.

## Figures and Tables

**Figure 1 nutrients-14-02125-f001:**
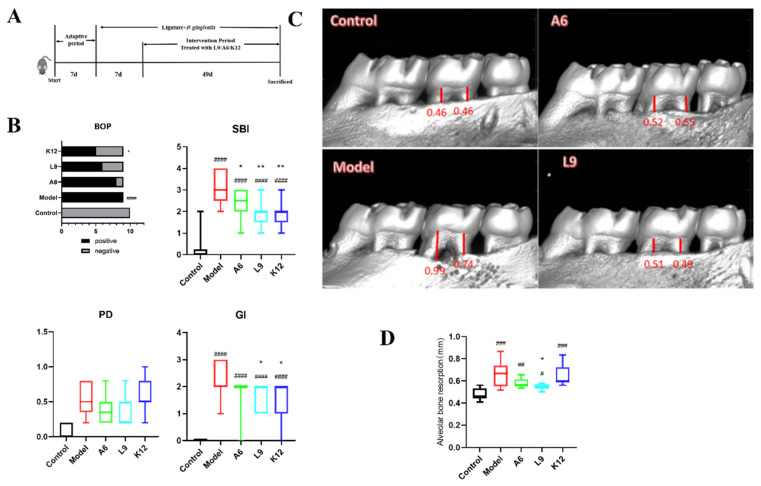
Effects of A6 or L9 administration on the clinical characterization of PD in mice from different groups. (**A**) Experimental arrangement. (**B**) Clinical index score—including bleeding upon probing (BOP), sulcus bleeding index (SBI), probing depth (PD), and gingival index (GI). (**C**) Representative micro-CT appearance of maxillary molars. (**D**) Micro-CT analysis of alveolar bone resorption in the maxillary second molars. Red bars represent the distance from the cementoenamel junction (CEJ) to the alveolar bone crest (ABC). Values designated with a superscript * reflect comparisons with the model group, and values with a # are in relation to the control group (** p* and *^#^ p* < 0.05; *** p* and *^##^ p* < 0.01; *^###^ p* < 0.001; ***** p* and *^####^ p* < 0.0001).

**Figure 2 nutrients-14-02125-f002:**
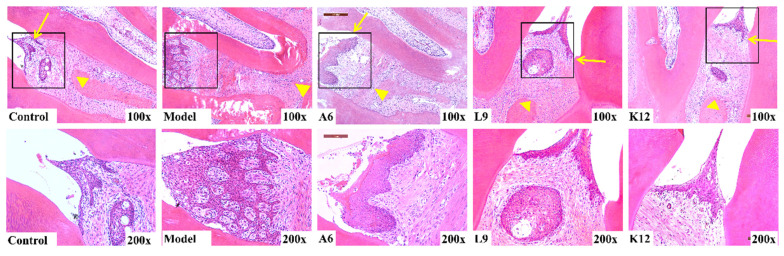
Effects of A6 or L9 administration on the histological changes in periodontal tissues in interproximal areas of the maxillary second molar (H&E staining, magnification, ×100 and ×200; yellow arrows represent CEJ, and yellow triangles represent ABC).

**Figure 3 nutrients-14-02125-f003:**
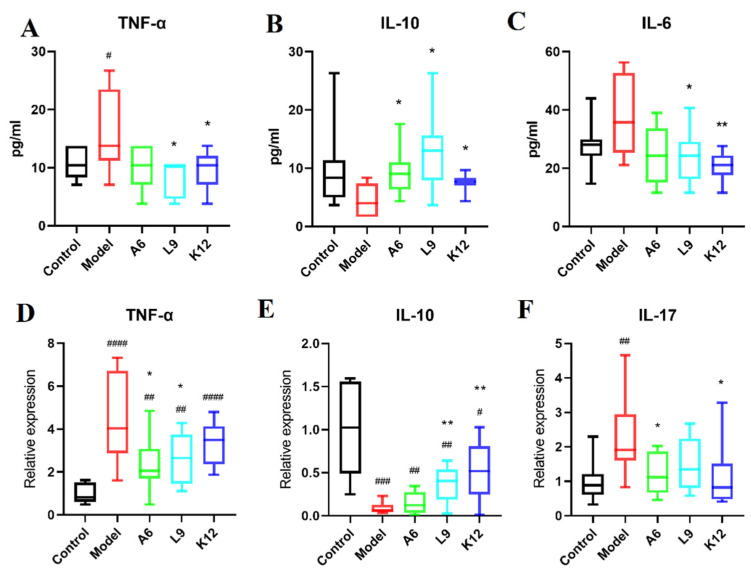
Reduced pro-inflammatory and enhanced anti-inflammatory cytokine expression in A6 or L9 intervention group. (**A**–**C**) TNF-α, IL-10, and IL-6 expression in serum as detected by ELISA. (**D**–**F**) Relative expression of TNF-α, IL-10, and IL-17 mRNAs in tissues as determined by RT-PCR. Values designated with a superscript * reflect comparisons with the model group, and values with a # are in relation to the control group (** p* and *^#^ p* < 0.05; *** p* and ^##^
*p* < 0.01; ^###^
*p* < 0.001; ^####^
*p* < 0.0001).

**Figure 4 nutrients-14-02125-f004:**
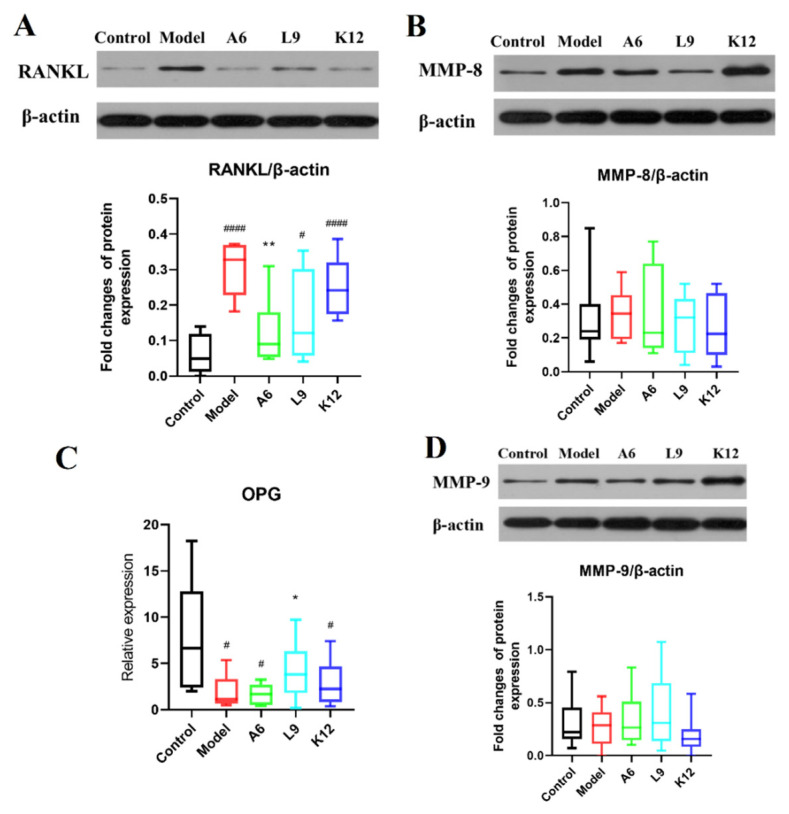
The modulating effect of A6 or L9 on protein expression levels of key biomarkers in gingival tissues. (**A**,**B**,**D**) Representative analysis and densitometric quantification of Western immunoblot bands. (**C**) Relative expression of OPG as determined by RT-PCR. Values designated with a superscript * reflect comparisons with the model group, and values with a # are in relation to the control group (** p* and *^#^ p* < 0.05; *** p* < 0.01; *^####^ p* < 0.0001).

**Figure 5 nutrients-14-02125-f005:**
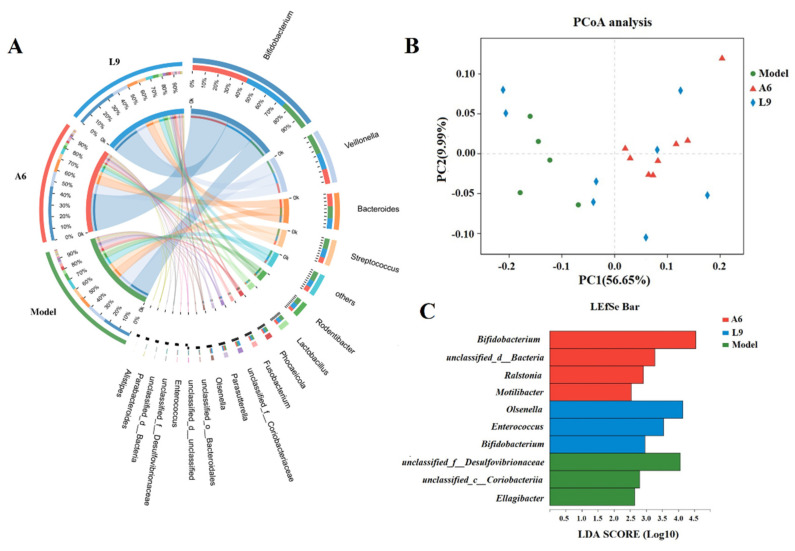
Taxonomic compositional changes to the subgingival microbiome after A6 or L9 intervention. (**A**) Circos diagram shows the taxonomic composition and the correspondences between samples and taxa of different groups at the genus level. (**B**) Principal coordinate analysis (PCoA) of different groups. (**C**) Linear discriminant analysis effect size (LEfSe) cladogram represents characteristic taxa associated with different treatments at the genus level.

**Figure 6 nutrients-14-02125-f006:**
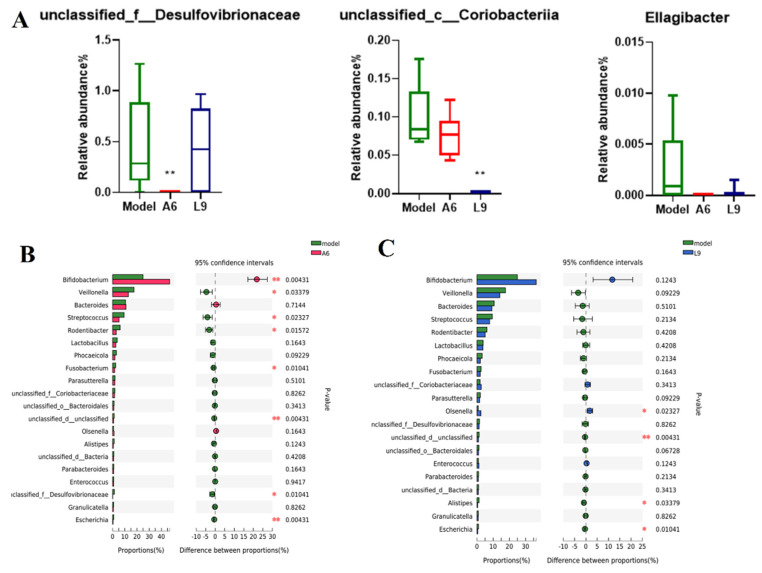
Effects of A6 and L9 administration on abundance changes in characteristic taxa at the genus level. (**A**) Characteristic taxa derived from LEfSe analysis. (**B**) Difference comparisons of species abundances between A6 and model groups using the Wilcoxon-Mann-Whitney U test. (**C**) Difference comparisons of species abundances between L9 and model groups using Wilcoxon rank-sum/Mann-Whitney U test (* *p* < 0.05, ** *p* < 0.01).

**Figure 7 nutrients-14-02125-f007:**
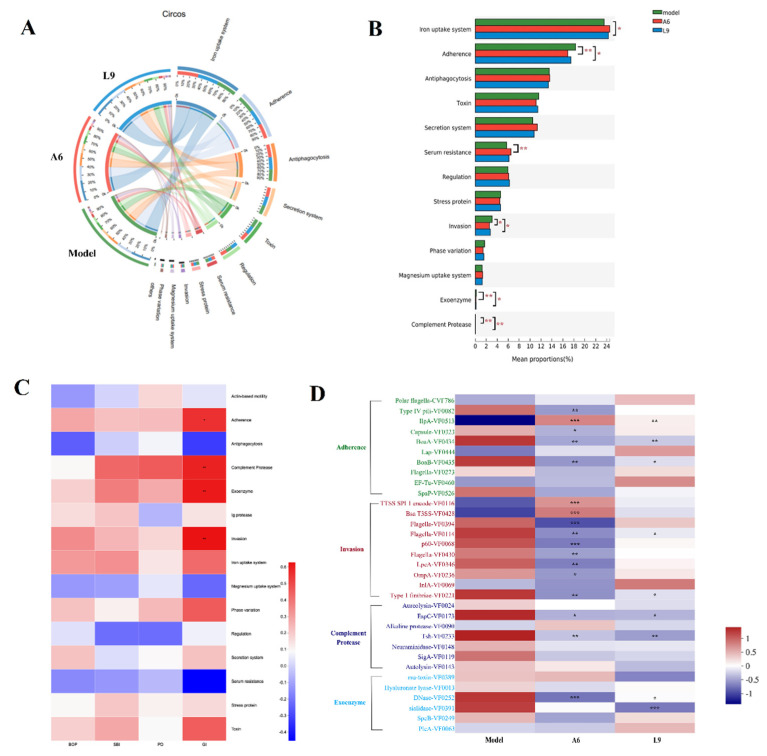
Effects of A6 or L9 administration on species virulence factor functions in PD-model mice. (**A**) The sample and species function Circos diagram shows the abundance correspondence between sample and virulence factor function. (**B**) Multigroup comparison diagram of virulence factor function depicting the significance of the functional abundances of virulence factors among different groups. (**C**) Correlation heatmap showing four clinical PD indices and species virulence factor functions. (**D**) Comparison of top 10 virulence factor differences in adherence, invasion, exoenzyme, and complement protease functions (* indicates the differences between probiotic and model groups; * *p* < 0.05, ** *p* < 0.01, *** *p* < 0.001).

**Table 1 nutrients-14-02125-t001:** Primer sequences.

Primer	Primer sequences(5′to3′)
TNF-α F′	TGAACTTCGGGGTGATCGGT
TNF-α R′	GGCTACGGGCTTGTCACTCG
IL10 F′	GCAGGACTTTAAGGGTTACTTGG
IL10 R′	TGCTCCACTGCCTTGCTTTT
IL6 F′	GATTGTATGAACAGCGATGATGC
IL6 R′	AGAAACGGAACTCCAGAAGACC
OPG F′	AGACGAGATTGAGAGAACGAGAA
OPG R′	ACGGTTTTGGGAAAGTGGTAT
GAPDH F′	TTCCTACCCCCAATGTATCCG
GAPDH R′	CCACCCTGTTGCTGTAGCCATA

## Data Availability

Not applicable.

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
