# Peer review of "Metagenomic Analysis Reveals a Mitigating Role for Lactobacillus paracasei and Bifidobacterium animalis in Experimental Periodontitis"

_nutrients, 2022, doi:10.3390/nu14102125_

Round 1

Reviewer 1 Report

The study was conducted with correct methodology, and the results are interesting. Being able to modify, in the future, the oral microbial flora into a more favorable one for periodontal health thanks to the introduction of probiotics in the diet, could be an effective prevention strategy on a large scale to reduce periodontal disease and its effects on oral health. .
However, the manuscript has greatly improved in drafting.
Only a few considerations and suggestions I would like to ask the authors
1) Report the limits of the study
2) Report the future implications of your findings

Author Response

Dear editors and reviewers:

    We appreciate all the efforts you have made to our manuscript. All the comments and criticisms from the reviewer are very valuable to us. We carefully replied to the comments of the reviewers and made corresponding modifications. The responses are numbered and the specific location of the modification is also mentioned in the reply for easy viewing. Thank you very much and looking forward to hearing from you soon.

The responses to the comments are as follows.

Comment: Report the limits of the study and the future implications of the findings.

Response: Thanks for the referee’s kind suggestion. According to your comments, we have added a relevant discussion of the experimental limits and the future implications of the finding in the last paragraph of the discussion.

Reviewer 2 Report

The authors aimed to investigate the ameliorating effect of Bifidobacterium animalis and  Lactobacillus paracasei on a ligature-induced PD rat model and to further verify their modifying effects on pro-inflammatory responses and the expression levels of key biomarkers in the experimental PD rat. I have several minor concerns regarding this article.

  1. You showed A6 and L9 can potentially decrease the amount of inflammation. But from a clinical perspective, since the oral microbiome is a complicated ecosystem, how would you specify their effect of them on other beneficial microbes in oral flora? How about the longevity of these microbial modifications?

  1. You used PD ligature model for periodontitis. Although it is a good model, particularly for assessing the regenerative treatment modalities, in your study, this model does not entirely match your aim. You needed a plaque-induced model for periodontitis to evaluate the effect of new bacteria on them.

  1. In terms of clinical evaluation, I can only see GI, SBI, BOP, and PD. However, you missed clinical attachment loss (CAL) which is a pivotal variable to assess the success of any periodontal therapies. Do you have any information to clarify that?

  1. Histological images (figure 2) in the result section do not indicate any clear results. Higher magnification with clear labelling is needed to correctly explain them.

Author Response

Dear editors and reviewers:

We appreciate all the efforts you have made to our manuscript. All the comments and criticisms from the reviewer are very valuable to us. We carefully replied to the comments of the reviewers and made corresponding modifications. The responses are numbered and the specific location of the modification is also mentioned in the reply for easy viewing. Thank you very much and looking forward to hearing from you soon.

The responses to the comments are as follows.

 Comment 1: A6 and L9 can potentially decrease the amount of inflammation. But from a clinical perspective, since the oral microbiome is a complicated ecosystem, how to specify their effect of them on other beneficial microbes in oral flora? How about the longevity of these microbial modifications?

Response: Thanks for the referee’s kind suggestion. In metagenomic sequencing analysis, we specifically focused on the taxa of a significant change after A6 and L9 intervention. However, no significant changes were found in other typical beneficial microbes in oral flora, except Bifidobacterium. The abundance of some typical pathogenic bacteria was significantly decreased. 

The longevity of these microbial modifications is also an issue of great concern to us. Previous relevant studies have not addressed this issue, but it directly affects the application of probiotics in the management of oral health. Therefore, based on the results of this study, we will design a new animal and human intervention experiment to investigate the longevity of these microbial modifications and the changes in relevant inflammatory indicators after stopping probiotic intervention.

Comment 2: Although PD ligature model is a good model for periodontitis., particularly for assessing the regenerative treatment modalities, in this study, this model does not entirely match our aim. We needed a plaque-induced model for periodontitis to evaluate the effect of new bacteria on them.

Response: Thank you very much for your professional suggestion. We very much agree with you that the simple PD ligation modeling method does not match our experimental purpose, we added Porphyromonas gingivalis smear infection to simulate the occurrence of periodontitis, which is mentioned in Line 130-135, page 4. And your idea is very consistent with us, we are also doing plaque culture in vitro for periodontitis modeling intervention.

Comment 3: In terms of clinical evaluation, only see GI, SBI, BOP, and PD. However, missed clinical attachment loss (CAL) which is a pivotal variable to assess the success of any periodontal therapies.

Response: Thanks for the referee’s kind suggestion. We agree with you very much. In terms of clinical evaluation, CAL is indeed a very important indicator. We also tried to evaluate CAL during the experiment. However, the teeth of experimental mice are too small, it is difficult to accurately find the enamel cementum boundary to evaluate CAL during exploration. Moreover, we found that in the relevant modeling methods, there was no research on the successful detection of CAL, so we gave up the evaluation of CAL.

 Comment 4: Histological images (figure 2) in the result section do not indicate any clear results. Higher magnification with clear labelling is needed to correctly explain them.

Response: Your suggestion is very important. We adopted your suggestion, supplemented and improved the histological result image, and added a higher magnification image in the revised manuscript.

This manuscript is a resubmission of an earlier submission. The following is a list of the peer review reports and author responses from that submission.

Round 1

Reviewer 1 Report

The article evaluates treating periodontal disease with the probiotic bacteria Bifidobacteria animalis A6 and Lactobacillus paracasei L9 using a rat model. While the article is detailed and descriptive, the need for extensive revision due to incorrect English grammar makes it difficult to completely and accurately evaluate the research and the author's conclusions. 

The methods were not adequately described. In particular, the description of the metagenomic sequencing and statistical methods used for analysis are very minimal and inadequate. Which Illumina machine was used for sequencing? Novaseq or Hiseq? There is not Novaseq/Hiseq hybrid.

More detailed methods and an overhaul of the writing for English grammar is needed. 

Author Response

Dear editors and reviewers:

We appreciate all the efforts you have made to our manuscript. All the comments and criticisms from the reviewer are very valuable to us. We have modified the manuscript carefully according to the comments of the reviewers, and all the changes were highlighted. The responses are numbered and the specific location of the modification is also mentioned in the reply to make your viewing easier.

Comment 1: The description of the metagenomic sequencing and statistical methods used for analysis are very minimal and inadequate, and not clear that the Illumina sequencing machine is Novaseq or Hiseq?

Response: Thanks for the referee’s kind suggestion. Your comment is very important, and we are aware of our misrepresentation. According to this advice, we supplemented our description on metagenomic sequencing, and we clarified the sequencing machine in this revised version, the sequencing was performed on Illumina NovaSeq(Illumina Inc., San Diego, CA, USA). The supplementary modification can be found in Line 196-219, pages 4-5.

Comment 2: Writing grammar issues need to be checked and revised.

Response: Thanks for the referee’s kind suggestion. We have carefully checked and revised the manuscript based on your comments.

Reviewer 2 Report

The study was conducted with correct methodology, and the results are interesting. Being able to modify, in the future, the oral microbial flora into a more favorable one for periodontal health thanks to the introduction of probiotics in the diet, could be an effective prevention strategy on a large scale to reduce periodontal disease and its effects on oral health. .
Just a few considerations and suggestions I would like to ask the authors

1) At present, there is no effective long-term prevention and treatment of PD. (I do not agree with this statement, if it is not necessary: remove it or reformulate it or give adequate bibliographic support to the statement).
2) Report the limits of the study
3) Report the future implications of your findings

Author Response

Dear editors and reviewers:

We appreciate all the efforts you have made to our manuscript. All the comments and criticisms from the reviewer are very valuable to us. We have modified the manuscript carefully according to the comments of the reviewers, and all the changes were highlighted. The responses are numbered and the specific location of the modification is also mentioned in the reply to make your viewing easier. 

Comment 1: Mentioned in the introduction “At present, there is no effective long-term prevention and treatment of PD.” is not necessary. Recommend remove it or give adequate bibliographic support to the statement.

Response: Thanks for the referee’s kind suggestion, and we have realized our mistake. The previous statement was indeed inaccurate, and we have deleted the sentence.

Comment 2: Report the limits of the study.

Response: Thanks for the referee’s kind suggestion. Your opinion is very important. The research object of this experiment is an animal model, which is different from the human body in terms of tooth structure, pathogenesis, and oral flora. The application of experimental strains requires further research. The application of experimental strains to improve human periodontitis requires further study. This is the limitation of this study, we have added in Line 562-565, page 14.

Comment 3: Report the future implications of the findings.

Response: Thanks for the referee’s kind suggestion. Your opinion is very important. We expect to provide evidence for the adjuvant treatment of periodontitis with probiotics, and to provide new alternative strains for oral health care products. This is the future implications of the findings, we have added in Line 559-562, page 14.